# Making the best of a bad sample: Comparison of DNA extraction and quantification methods using sub-optimally stored *Ixodes ricinus* ticks

**Andrea P. Cotes-Perdomo**[1,2*], **Kenedith Méndez-Gutierrez**[3], **Kristian Alfsnes**[4],
**Åshild K. Andreassen**[1,2], **Andrew Jenkins**[1]

**1** Department of Natural Science and Environmental Health, University of South-Eastern Norway, Bø, Norway, **2** Department of Virology, Norwegian Institute of Public Health, Oslo, Norway, **3** Faculty of Basic Sciences, Universidad del Magdalena, Santa Marta, Colombia, **4** Department of Bacteriology, Norwegian Institute of Public Health, Oslo, Norway

* aperd@usn.no

## Abstract

The primary aim of this study was to evaluate the performance of four DNA extraction methods on a collection of 200 sub-optimally stored *Ixodes ricinus* ticks from Southeastern Norway (177 nymphs, 11 males and 12 females). The methods were ammoniium hydroxide hydrolysis of homogenised or intact ticks and two commercial silica membrane-based kits, the QIAGEN Blood and Tissue kit and the QIAGEN Mini kit. DNA was evaluated by spectrophotometry, fluorometry, agarose gel electrophoresis and quantitative PCR. The second aim was to compare methods of evaluating the yield and purity. All four extraction methods provided amplifiable DNA, but the yield was variable (median 151 ng, range 0.13 to 4500 ng). DNA yields were not significantly different across the methods. Nine of 200 samples were inhibitory, all of which were ammonium hydroxide extracts of homogenised ticks. DNA purity, as judged by $A_{260/280}$ ratios, was low; it was highest (mean = 1.63) for the Qiagen Blood and Tissue kit with the other methods showing values averaging 1.44. DNA yield measurements using qPCR, fluorometry (Qubit), drop spectrophotometry (NanoDrop) and gel electrophoresis (E-gels) correlated poorly, *r* ranging from <0 to 0.9 with no systematic pattern (average 0.4), probably reflecting the effects of low purity, low concentrations and differing amounts of single- and double-stranded DNA. *Neoehrlichia mikurensis* and *Borrelia burgdorferi* s.l. were detected in 5 and 13% of samples, respectively by qPCR. Our findings indicate that for the purposes of qPCR analysis, ammoniium hydroxide hydrolysis of intact ticks, a very cheap and simple method is as good as any of the other methods tested.

**Data availability statement:** All relevant data are within the paper and its Supporting Information files.

**Funding:** This work was in part financed by The Norwegian Lyme Borrelioses society (NLBF). The funders had no role in study design, data collection and analysis, decision to publish, or preparation of the manuscript.

**Competing interests:** The authors have declared that no competing interests exist.

## Introduction

Ticks are hematophagous parasites with a large range of vertebrates hosts which serve as vectors for a wide range of pathogens and are thus of major importance for human and animal health [1]. Most of the pathogens are difficult or impossible to culture and their detection relies on molecular methods, principally PCR amplification, supported by sequencing for identification and confirmation. [2]. The reliability of such analyses is dependent upon DNA of adequate quantity and quality, particularly when next-generation sequencing is to be used [3,4], and a trustworthy quantification process is needed. Yield and quality are impacted by the nature of the sample and the method of extraction [5,6]. The extraction method must efficiently release DNA from the ticks' tough chitinous exoskeleton and remove potential inhibitors of downstream analyses. A further issue is DNA degradation, which may occur if ticks are stored for long periods under non-optimal conditions [7,8]. Thus, ticks are usually stored at -20 and/or in ethanol whose antibacterial effect prevent microbial degradation and whose protein denaturant properties will inhibit nucleases. [9]. Inadequate DNA may result in failure to detect pathogens at low pathogen load, which in turn will lead to underestimation of pathogen prevalence.

Several DNA extraction methods, involving various combinations of physical, chemical and enzymatic disruption and lysis have been developed and adapted for use with ticks [10,11]. Many employ commercially available kits, which offer the advantage of ready-made, quality-controlled reagents, standardized, simple to follow protocols, and the potential for automation, but which may be relatively expensive and, when performed manually, labour-intensive. Typically, such methods involve sample lysis with enzymes and detergents followed by selective absorption and elution of nucleic acids on silica membranes. Such multi-step procedures introduce the possibility of operator error. For eco-epidemiological studies which may require thousands of ticks, the price of kits may become a significant part of the project budget, potentially a prohibitive one. Finally, the composition of the reagents is proprietary information, which may cause reproducibility issues if the kit becomes unavailable. In-house methods lack the advantages of kit-based methods but are usually much cheaper and fully traceable. The ammonium hydroxide hydrolysis method [12–14] has the advantage of being both simple and cheap as it uses only one reagent, ammonium hydroxide, whose price is negligible. The product is a crude lysate containing denatured DNA and more-or-less hydrolysed protein and other components of the tick.

Various methods for measuring the concentration and quality of DNA are available. The standard method is determination of spectrophotometric absorbance at 260 and 280 nm. This measures the absorbance peak of the pyrimidine ring at 260 nm, while the ratio of $A_{260}$ to $A_{280}$ is a measure of DNA purity, pure single-stranded DNA having an $A_{260/A280}$ ratio of 1.8 to 2.0. Lower ratios are considered indicative of protein contamination, as protein, while it also absorbs at 260 nm, has stronger absorbance at 280 nm. Modern instruments, such as the Nano Drop used in this study can perform measurements on 1μl samples (drop spectrophotometry).

While this method is quite accurate for pure double-stranded (DS) or single-stranded (SS) DNA and corrections for the presence of moderate amounts of protein are available, $A_{260}$ will not be an accurate measure of DNA concentration if the sample contains a mixture of DS and SS DNA (whose $A_{260}$s differ) or if other substances containing aromatic rings, such as ATP, RNA, and phenol are present. Further, the $A_{260/280}$ ratio is not a foolproof measure of molecular purity as many impurities have no absorbance in this region. It would also be possible to construct a mixture of aromatic compounds having an $A_{260/280}$ ratio of 1.8 while containing no DNA at all.

Fluorometric methods using DNA binding dyes provide more specific tests and have been developed into commercial tests like the Qubit instrument used in this study. Such tests can quantify SS or DS DNA, depending on the dye used, but not both. However, fluorescence is affected by the base composition of the DNA and for the best accuracy the instrument needs to be calibrated against homologous DNA of known concentration. A second issue is the phenomenon of quenching, where the fluorescence excitation energy is transferred to a second molecule before fluorescence can occur, depending on whether such molecules are present as impurities in the sample.

Ad hoc fluorescence methods have been in use in molecular biology laboratories since the invention of ethidium bromide-stained agarose gels [15]. They are typically used semi-quantitatively, either after gel electrophoresis or by directly spotting DNA onto agarose containing a fluorescent dye, though if combined with scanning and integrating the fluorescence signal, they may be fully quantitative, albeit not very precise. Sensitivity to DS DNA, SS DNA and RNA vary depending on the dye used. If electrophoresis is used the method has the advantage of simultaneously providing a measure of the integrity of the DNA.

Quantitative real-time PCR (qPCR) offers a third approach to DNA quantification. qPCR has a much greater dynamic range than the methods described above and is capable of quantifying amounts of DNA that are undetectable by such methods. qPCR only quantifies the target genome. Due to the exponential dynamics of the PCR reaction, qPCR has relatively low precision. Inhibitors and quenchers can cause inaccurate results, but the anomalous amplification curves that they cause are easily identified [16].

In this study we compare the performance of two related commercial DNA extraction kits that use silica membranes with that of two variants of the ammonium hydroxide hydrolysis method using a challenging sample material: a collection of questing *I. ricinus* ticks that had been subjected to long-term storage under sub-optimal conditions, at 4°C without full ethanol immersion. We test the viability of the material obtained to detect naturally acquired tick-borne bacteria. Inhibition was detected by the spiking method, in which a small quantity of target DNA is added together with an aliquot of the sample and a qPCR test is performed [17]. If the sample is inhibitory, qPCR efficiency is impaired and the Cq value is increased relative to controls and non-inhibitory samples [18]. In addition, we compare the applicability of drop spectrophotometry, fluorometry, gel-electrophoresis and qPCR for assessing the quality and quantity of the DNA extracts.

## Materials and methods

### Ethics

Ticks were collected in open forests with full public access under Norwegian law. No permit was required for this work.

### Collection of samples and DNA extraction

Ticks were collected in the localities of Asdal (59.07°N, 9.60°E: 2 females, 6 males and 12 nymphs), Langøya (59.01°N, 9.75°E: 10 females, 5 males and 4 nymphs), and Jomfruland (58.88°N, 9.61°E: 1 male and 160 nymphs) in Telemark, Southeastern Norway by flag-dragging in the spring and summer of 2011. These samples were preserved in absolute ethanol at 4°C; by the time of extraction (October 2022) the ethanol had evaporated completely in more than 90% of the samples, the remaining samples were partially submerged in ethanol.

DNA was extracted from individual ticks by four different methods as described below. Two-hundred questing *I. ricinus* ticks were included: 12 females, 11 males, and 177 nymphs (Table 1). For each method, 50 ticks were used, including 3

**Table 1. Number of samples and methods of DNA extraction.**

| Method | Homogenisation | Code | Females | Males | Nymphs | Total |
|---|---|---|---|---|---|---|
| Ammonium hydroxide | No | ANC[a] | 3 | 3 | 44 | 50 |
| Ammonium hydroxide | Yes | AC[b] | 3 | 3 | 44 | 50 |
| Qiagen Mini Kit | Yes | QMK | 3 | 3 | 44 | 50 |
| Qiagen Blood and Tissue Kit | Yes | QBT | 3 | 2 | 45 | 50 |

[a]Ammonia, not crushed.

[b]Ammonia, crushed.

females, 3 males and 44 nymphs, except for the Qiagen blood and tissue kit, where only two males were available, and an extra nymph was substituted to bring the number to 50. The volume of DNA extract was 100 µl in all methods. Ammonia extraction methods were based on the method described by Guy and Stanek [12].

[1] **Qiagen Mini Kit** (QMK) (Qiagen, Hilden, Germany). Individual ticks were crushed by agitating them in a 2 ml tube containing 180 µl of the lysis buffer provided (RLT) and three 2.5 mm stainless steel beads for 5 minutes at 10 000 Hz using a Bead Bug microtube homogenizer (Benchmark Scientific, Sayreville, NJ. USA). DNA was extracted following the manufacturers' instructions except that the lysis step was extended to 16 hours, and the DNA was eluted from the silica membrane in two 50 µl portions to bring the final elution volume to 100 µl.

[2] **Qiagen Blood and Tissue kit** (QBT). As (1) above, except the Qiagen Blood and Tissue kit (Qiagen®, DNeasy®) was used following the manufacturer's instructions.

[3] **Ammonium hydroxide extraction of intact ticks** (ANC – "Ammonia, non-crushed"). Individual ticks were placed intact in a 1.5 ml Eppendorf tube containing 100 µl 2.5% $NH_4OH$, with a small hole pierced in the lid with a syringe needle to prevent pressure buildup, and incubated at 99°C for 20 minutes, after which the lid was opened, and the tubes were incubated for a further 10 minutes at the same temperature to evaporate the ammonia. Samples were then stored at 4°C for one week to allow residual ammonia to escape.

[4] **Ammonium hydroxide extraction of homogenized ticks** (AC – "Ammonia, crushed"). As described above (3), but prior to ammonium hydroxide hydrolysis at 99°C the individual ticks were crushed as in (1) but in a 2ml tube containing 100 µl 2.5% $NH_4OH$.

## Assessment of removal of ammonia

In order to assess successful removal of ammonia after extraction, the pH indicatory dye Neutral Red was added to the ammonium hydroxide solution at a concentration of 200 µg/ml and extraction of intact ticks was conducted as in (1) above. The colour of the samples was observed immediately after the evaporation stage. The tubes were then closed, and the samples were placed at 4°C and observed at daily intervals. The pierced cap was not sealed. The ticks used in this experiment were not included in the main study as Neutral Red was found to inhibit the PCR reaction.

## Estimation of DNA concentration and purity

Drop spectrophotometry at 260 and 280 nm was performed using the Nanodrop Lite Spectrophotometer (Thermo Scientific, Wilmington DE, USA). Two measurements were made, using the instrument settings for DS and SS DNA, respectively. For each measurement, $A_{260}$, $A_{280}$, $A_{260}/A_{280}$ ratios and the instrument-derived DNA concentration were recorded. 1 µl of DNA was used for each assay.

Fluorometric measurement of DS DNA concentration was performed in a Qubit 3.0 fluorometer (Thermo Fisher Scientific, Wilmington, DE, USA), using the Qubit dsDNA HS (High Sensitivity) assay kit according to the manufacturer's instructions, adding 1µl of DNA.

Gel electrophoresis was performed on 4% agarose gels stained with SYBR-green using the Invitrogen E-Gel apparatus (Invitrogen, Waltham MA, USA). As a standard, 0.5 µg of the Invitrogen 1kb Plus DNA Ladder was used. 18 µl of each sample were added plus 2 µl of E-Gel™ sample loading buffer 1X. Runs lasted 15 minutes at 48 V. Gels were photographed with the instrument's inbuilt camera. For quantitative analysis, E-gel photographs were downloaded and converted to flat, negative greyscale images using Adobe Photoshop. These were imported into Quantity One 4.6.3 gel analysis software (BioRad Laboratories, Hercules, CA, USA). DNA smears were integrated using the 'Volume' analysis tool. For each lane, a rectangle containing the visible DNA smear was selected, and its summed intensity ('volume') was assessed and corrected for background using the 'local background subtraction' setting. The corrected 'volume' (V) was thus the sum of the intensity of each pixel within the rectangle minus the mean intensity of pixels immediately adjacent to the rectangle: $\Sigma$ (smear pixel intensity – mean background pixel intensity). Smear volume was converted to ng DNA by reference to the volume of the standard DNA ladder (500ng): $V_{smear}/V_{ladder} \times 500 = DNA_{smear}$ (ng). Where no smear was visible to the naked eye, or where background fluorescence exceeded sample fluorescence, resulting in a negative value, a value of zero was recorded.

Quantitative real-time PCR targeting a 150 bp fragment of the *I. ricinus 5.8S* rRNA, a small RNA found in the large subunit of eukaryotic chromosomes, was performed on the StepOne Real-Time PCR System (Thermo Fisher Scientific) using Power UpTM SYBR™ Green Master Mix, following the method described in Tables 2 and 3 [19,20]. 5µl of DNA were added. A 10x dilution series of *I. ricinus* DNA of known concentration was used to construct standard curves for quantitation ( and 3). Melting curves were checked to confirm that the sample melting peak coincided with that of the controls.

Quantitative PCR was used as the reference measure of DNA concentration because it has a greater dynamic range than spectrophotometric or fluorometric methods, detects DS and SS DNA equally well and because any inhibitory effect of other materials in the sample will be evident as an anomalous amplification curve [16].

For all methods, yield estimate distributions are expressed as median and range as a high degree of skew invalidated the use of mean and standard deviation.

**Table 2. Sequences of primers (*5.8S*) used in the *I. ricinus* PCR assay.**

|  | Sequence (5'-3') | Reference |
|---|---|---|
| Forward | GGAAATCCCGTCGCACG | [20] |
| Reverse | CAAACGCGCCAACGAAC | |

**Table 3. PCR conditions for the *I. ricinus* PCR assay.**

|  |  | Temp.°C | Time |
|---|---|---|---|
| **Pre-cycling stage** | Step 1 | 50 | 2 min |
|  | Step 2 | 95 | 10 min |
| **Cycling stage (45 cycles)** | Step 1 | 95 | 15 sec |
|  | Step 2 | 60 | 1 min |
| **Melt curve stage** | Step 1 | 95 | 15 sec |
|  | Step 2 | 60 | 1 min |
|  | Step 3 | 60–95, 0.3°C increments | |
| **Holding stage** | – | 15 | 1 min |

## Assessment of PCR inhibition

To detect PCR inhibition, all samples were spiked with $6 \times 10^5$ copies of plasmid pNeo containing the *groEL* target sequence [22], and *N. mikurensis* real time PCR was performed. 5µl of each extract DNA were added. The threshold line was set to 20 000 relative light units (RLU). Samples that had increased Ct values were considered inhibitory.

## qPCR for detection of *Neoehrlichia mikurensis* and *Borrelia burgdorferi sensu* lato

*Neoehrlichia mikurensis* was detected by SYBR-green real time PCR targeting the *groEL* gene and *B. burgdorferi* s.l. was detected by real time PCR targeting the *flaB* gene fragment as previously described [21,22], adding 5µl of DNA.

## Statistical analyses

For all statistical analyses, DNA concentrations were converted to total yield in ng.

A Kruskall-Wallis test was performed for each of the quantity and quality measurements. The quantity of DS and SS DNA expressed in ng from both NanoDrop and Qubit, the $A_{260}$ absorbance and $A_{260/280}$ ratio, the genome copies number for *I. ricinus* and the Ct values for *N. mikurensis* spiked samples. When significant differences (<0.05) between extraction methods were found, a post-hoc Mann-Whitney-Wilcoxon tests with the Bonferroni P-value adjustment was done.

To check the influence of DNA quality variables in qPCR quantitative measurement, a generalized linear model (GLM) using Gamma distribution with a log link function, as the response variable was positively skewed, was fit. The full model included the $A_{260}$ absorbance, $A_{260/280}$ ratio, the Ct values of the spiked samples with *N. mikurensis* DNA and their interactions with each extraction method. Model simplification was performed using a backward stepwise selection based on the Akaike information criterion, it was implemented using the *stepAIC* function from the *MASS* package in R.

A Bland-Altmann analysis was performed presenting the Qubit/qPCR vs NanoDrop/qPCR ratios employing the qPCR quantification as the expected DNA yield. The ratios were obtained dividing each Qubit and NanoDrop measurement by its parallel measurement by qPCR in ng. The media was calculated between Qubit and qPCR and between Nano and qPCR measurements.

Pearson's correlation coefficient analysis was performed to evaluate the correspondence and proportionality between DNA yield measurements, calculated in Microsoft Excel using the CORREL function.

R version 4.3.3 was used for all statistical analysis and generation of figures, except for Pearson's correlation coefficients.

# Results

## Change in pH during and after extraction

The successful removal of ammonia in the evaporation stage was assessed by observing the colour change of the pH indicator Neutral Red. Neutral Red is red below pH 6.8, yellow above pH 8.0 and grades through shades of orange at intermediate pH.

Immediately after the evaporation stage, all samples were clear yellow, indicating that the pH remained above 8.0, although no odour of ammonia was detectable, and a faint orange tinge was observable in some samples. After two days' storage at 4°C, some samples had developed a distinct orange colour, while others remained yellow. After five days, all samples were orange, indicating near-neutral pH.

## Quantitation and quality of DNA

**DNA quantity/yields.** Drop spectrophotometry reported positive values for all samples. The median yield estimate was 680 ng with values ranging from 40 to 3600 ng for SS DNA, and 1180 ng with values ranging from 280 to 19700 ng for DS DNA. The means were 875 ng and 1804 ng for SS and DS DNA, respectively.

Fluorometry reported DS DNA median yield estimate of 88 ng with values ranging from 12 to 1260 ng. The mean was 162 ng.

For 16 samples, the fluorometric value was below the limit of quantification and was recorded as zero. For these samples, the median yield estimates were 16 ng by qPCR and 510 ng and 1310 ng by spectrophotometry for SS and DS DNA, respectively. By gel electrophoresis, 10/16 of these samples showed no signal, leading to a median of zero; the mean, including the zero values, was 3 ng.

When samples were analysed by gel electrophoresis, 51 samples showed no visible DNA. Sixteen further samples gave negative results due to excessive background and were excluded from quantitative analysis. For the remaining 133 samples, the median estimated DNA yield was 43 ng, with values ranging from 5 to 4370 ng. The mean was 367 ng.

All samples across the different DNA extraction methods yielded amplifiable *I. ricinus* DNA, as judged by qPCR. The median yield estimate was 151 ng, with values ranging from 0.13 to 4500 ng; the mean was 385 ng. Four samples gave estimated yields of less than 1 ng; eleven samples gave estimated yields of less than 10 ng. The yield distribution is shown in Fig 1.

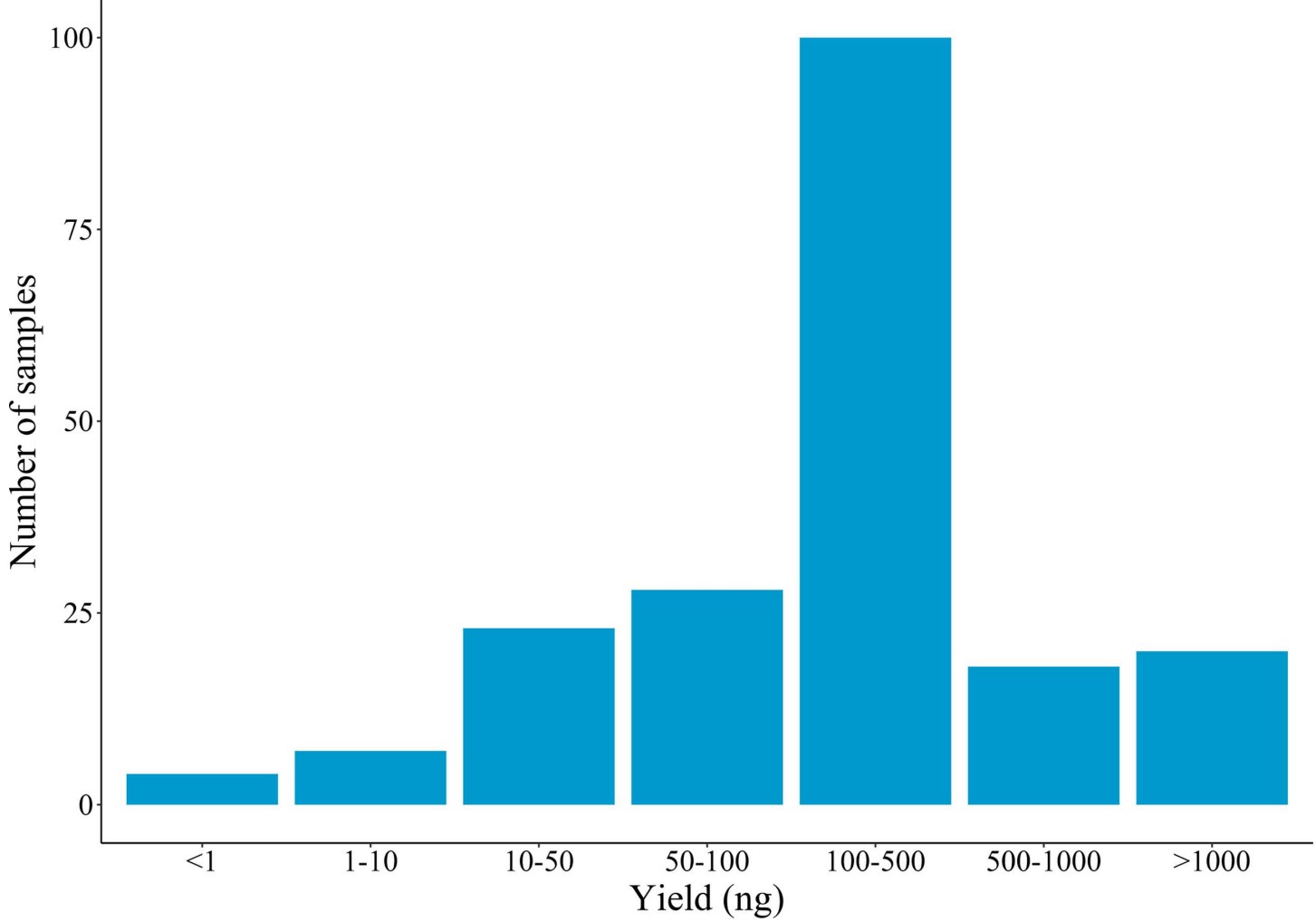

**Fig 1. Distribution of DNA yields as estimated by qPCR.**

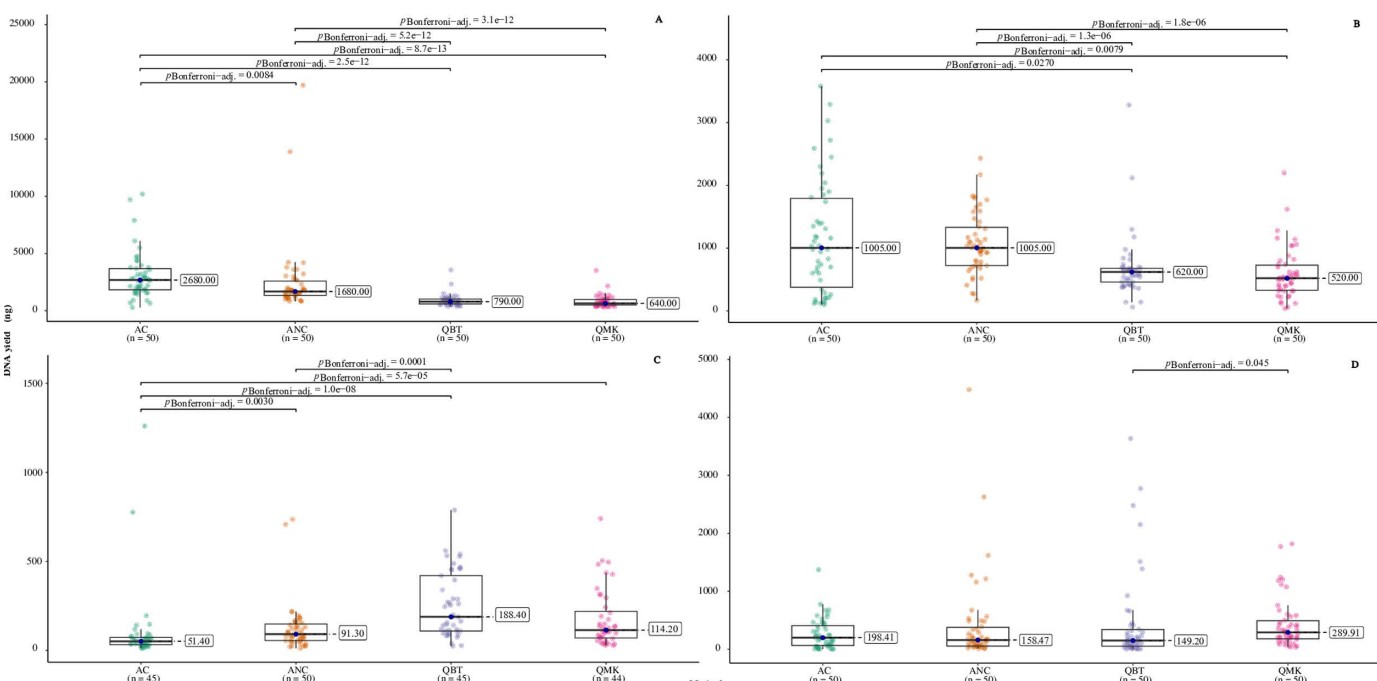

**Fig 2. Yield distribution for the four extraction methods as estimated by spectrophotometry (NanoDrop) for DS (A) and SS DNA (B); fluorometry (Qubit) (C); and qPCR (D).** AC: Ammonium hydroxide, homogenised ticks; ANC: Ammonium hydroxide, intact ticks; QBT: Qiagen Blood and Tissue Kit; QMK: Qiagen Mini Kit.

As judged by qPCR, median yields were similar (149–198 ng) for the two ammonia-based extractions and the Qiagen blood and tissue kit, but higher (299 ng) for the Qiagen mini kit (Fig 2D; Table 5). The difference was significant only between the two Qiagen kits but was also seen in the spectrophotometric and fluorometric measurements (Fig 2A, B, C; Table 4).

Spectrophotometry reported significantly higher yields of DNA for the ammonia-based methods, while fluorometry indicated the reverse (Fig 2A, B, C; Table 4). These differences are probably experimental artefacts, as discussed below.

**DNA quality.** Three measures of DNA quality were used: The 260/280 nm absorbance ratio for purity, gel electrophoresis for DNA integrity, and spiking analysis for PCR inhibition.

The mean $A_{260/280}$ ratios for the AC and QMK methods were 1.42 and 1.43, respectively, while the ANC and QBT methods gave significantly higher mean ratios: 1.49 and 1.63, respectively (Fig 3A; Table 4). Only four samples showed $A_{260/280}$ ratios in the range 1.8–2.0 indicative of pure DNA. These were extracted with the Qiagen Blood and Tissue (N = 3) or Qiagen Mini (N = 1) kits.

The electrophoretic patterns of samples extracted with Qiagen kits and ammonia hydroxide hydrolysis differed strikingly (Fig 4, Table 5, S1-S20 Figs). Qiagen extracts typically showed a compact band-like peak at high molecular weight (>15 Kb) with a more-or-less conspicuous tail of lower-molecular weight material. This pattern was seen in 64/100 samples; the remaining samples showed either no detectable DNA (23/100) or a flat distribution reminiscent of that seen for ammonium hydroxide hydrolysis samples (12/100), which may, in this case, suggest partial degradation. One sample showed a peculiar, multimodal pattern not observed in any other samples (S7 Fig). The typical pattern for ammonia hydroxide hydrolysis samples was a flat distribution beginning at very high molecular weight and extending to below 1000 bp. This pattern was seen in 73/100 samples, the remaining samples showing no visible DNA (25/100) or a diffuse band of lower molecular weight material, indicative of degradation (2/100). In summary, high molecular weight material was detectable in 77% of

**Table 4. P values (Bonferroni adjusted) from group comparison by Kruskal-Wallis and pairwise comparisons by Mann-Whitney-Wilcoxon tests. Significant results in bold type.**

| Measurement | Kruskal-Wallis | Mann-Whitney-Wilcoxon | | | | | |
|---|---|---|---|---|---|---|---|
| | | AC/ ANC | AC/ QBT | AC/ QMK | ANC/ QBT | ANC/ QMK | QBT/ QMK |
| **Qubit** | 0.0000 | **0.0003** | **0.0000** | **0.0000** | **0.0000** | 0.1636 | 0.1319 |
| **NanoDrop SS** | 0.0000 | 1 | 0.0660 | **0.0280** | **0.0000** | **0.0000** | 1 |
| **NanoDrop DS** | 0.0000 | **0.0016** | **0.0000** | **0.0000** | **0.0000** | **0.0000** | 0.7068 |
| ***I.r* 5.8S Ct** | 0.0000 | 1 | **0.0044** | 0.2454 | **0.0000** | **0.0199S** | 0.6426 |
| **$A_{260}$** | 0.0000 | 1 | **0.0000** | **0.0000** | **0.0000** | **0.0000** | 0.6600 |
| **$A_{260/280}$** | 0.0000 | **0.0195** | **0.0000** | 1 | 0.0555 | 1 | **0.0017** |
| ***N.m* Spike Ct** | 0.0000 | **0.0000** | **0.0022** | **0.0000** | 0.1143 | 1 | 0.6318 |

*I.r: Ixodes ricinus*; *N.m: Neoehrlichia mikurensis*.

**Table 5. Electrophoretic patterns by method. Peaked (clear compact band of high molecular weight); flat (from high to low molecular weight); peculiar (multimodal); degraded (diffuse and low molecular weight), non (no detectable DNA).**

| Extraction method | Peaked | Flat | Peculiar | Degraded | None |
|---|---|---|---|---|---|
| **Ammonia** | – | 73 | – | 2 | 25 |
| **Qiagen** | 64 | 12 | 1 | . | 23 |

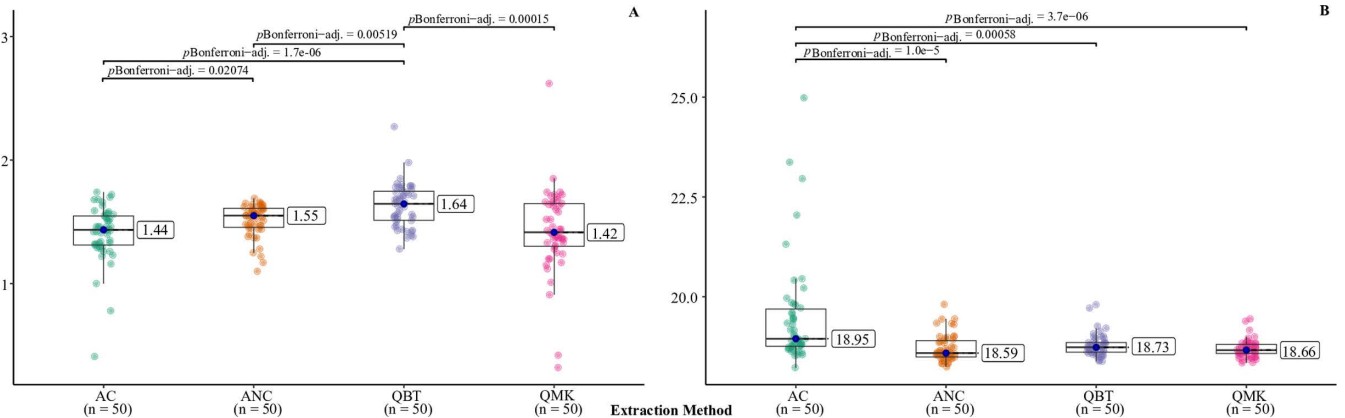

**Fig 3. Comparison of DNA quality between the four extraction methods based on the $A_{260/280}$ ratio by NanoDrop (A) and PCR inhibition test (B).** AC: Ammonia, homogenised ticks; ANC: Ammonia, intact ticks; QBT: Qiagen Blood and Tissue Kit; QMK: Qiagen Mini Kit. Results from the Kruskal-Wallis test are shown and significant (p < 0.05) differences in pair-wise comparison by Wilcoxon-Mann-Whitney with Bonferroni adjustment are shown at the top.

Qiagen extracts and 73% of ammonia extracts; 23% of Qiagen extracts and 25% of ammonia extracts had no detectable DNA, and 2% of ammonia extracts and 12% of Qiagen extracts showed patterns suggestive of degradation (see Table 5).

The results of spiking indicated that none of the samples extracted using the Qiagen kits or ammonia extraction of intact ticks were inhibitory. Ct values were in the range 18.2–19.9 and there was no significant difference between these methods (Fig 3B; Table 4). For ammonia extracts of homogenised ticks, nine samples showed raised Ct values, in the

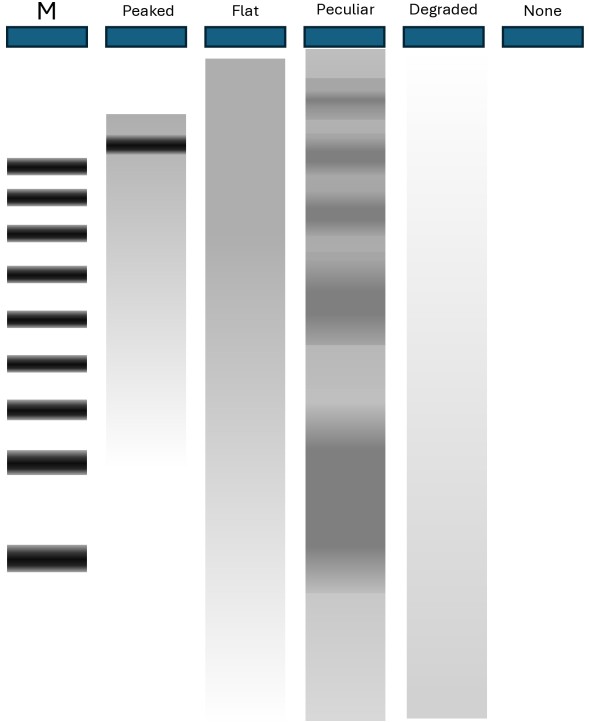

**Fig 4. Summary of electrophoresis patterns of tick DNA extracted with Qiagen kits (QBT, QMK) and ammonia (AC, ANC) as described in Table 5.** M: standard.

range 19.9–25, indicating slight to moderate inhibition, while the remaining 41 samples showed normal values. The mean Ct value for this method was thus significantly higher (Fig 3B; Table 4).

The final GLM included as predictors the extraction method, the $A_{260}$ absorbance, the Ct of the spiked samples, and the interaction between the Ct of the spiked samples and each extraction method. The ANC method had a significant negative effect on the qPCR estimated DNA yield (Est = -37.08, p-values = 0.0005), while the effect of the other methods was not statistically significant (QBT: Est = 3.74 and p-values = 0.76; QMK: Est = -4.03 and p-values = 0.80). The main effect of the $A_{260}$ absorbance was negative and almost significant (Est = -1.08, p-value = 0.068). The Ct value of spiked samples did not have a significant main effect (Est = -0.027, p-value = 0.844). But a significant interaction was observed between this Ct value and the ANC method (Est = 1.99, p-value = 0.0005). This interaction was not significant for the other extraction methods (QBT: Est = -0.14 and p-values = 0.83; QMK: Est = 0.37 and p-values = 0.65).

### Comparison of measurement methods

The ratios of DNA yield estimates were obtained by dividing theQubit and NanoDrop estimates bythe corresponding qPCR estimate, and the median ratios were then detemined for each extraction method. Relative to qPCR, spectrophotometry overestimated DNA yield, while fluorometry underestimated it except for Qiagen blood and tissue extracts (Table 6, Fig 5).

Correlation between the DNA measurement methods is shown in Table 7. Correlation coefficients, although mostly positive, varied between -0.11 and 0.97, indicating low covariance among quantifications obtained by different technologies. Although higher correlation coefficients were expected where the DNA was DS (QIAGEN) and the measurement method was optimized for DS DNA, this was not found to be the case.

**Table 6. Comparison of DNA measurement methods for the different extraction methods.** The table shows median ratios of yield estimate relative to qPCR for fluorometric detection of double-stranded DNA (Qubit) and spectrophotometry (NanoDrop) with settings for double-stranded (DS) and single-stranded (SS) DNA. Values <1 indicate underestimation of DNA yield while values >1 indicate overestimation.

| | Measurement method | | |
|---|---|---|---|
| **Extraction method** | **Qubit** | **NanoDrop DS** | **NanoDrop SS** |
| **Ammonia, intact** | 0.5 | 12 | 7 |
| **Ammonia, homogenized** | 0.2 | | |
| **Qiagen Mini** | 0.5 | 2.5 | NA |
| **Qiagen Blood & Tissue** | 1 | 5 | NA |

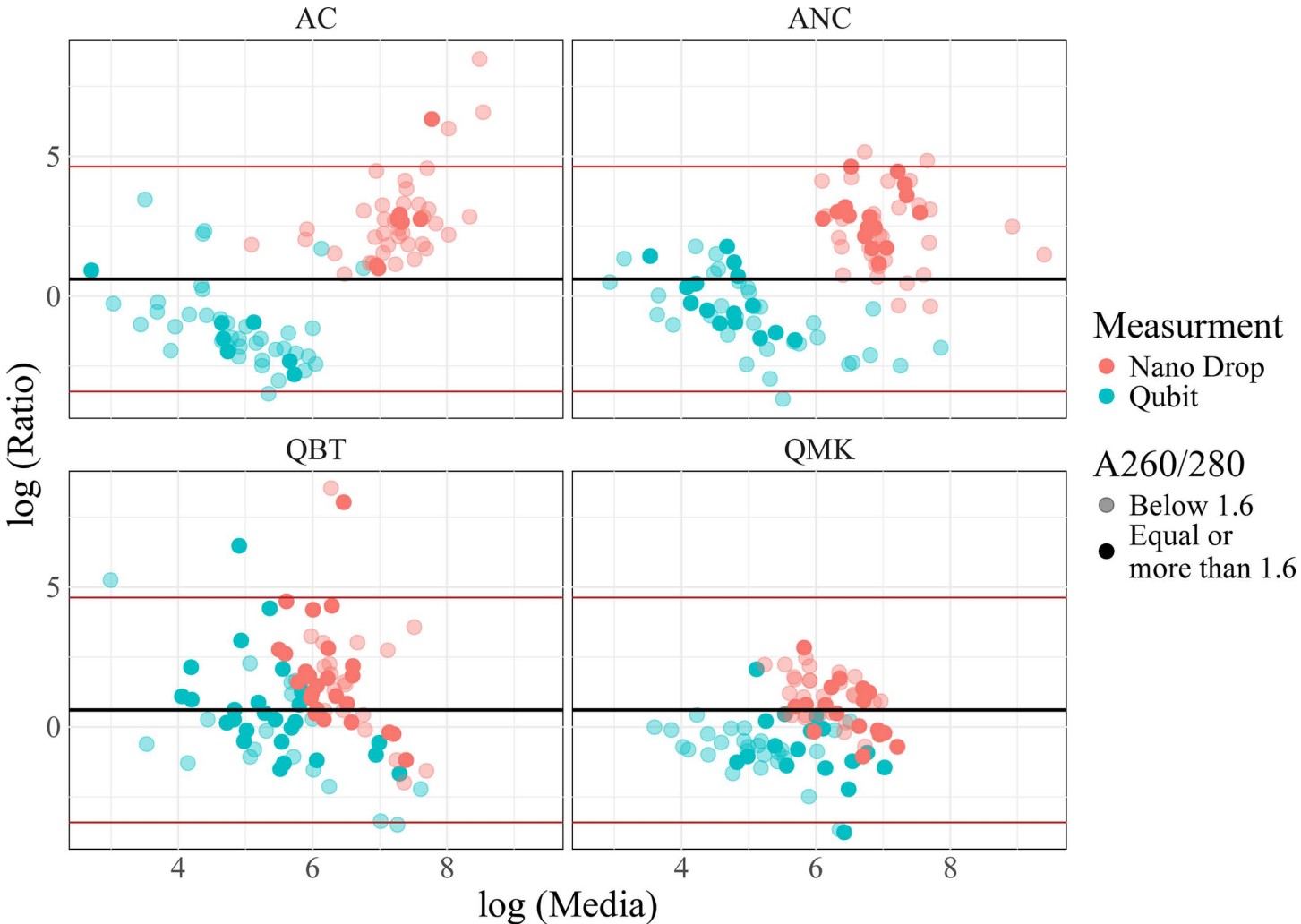

**Fig 5. Bland-Altmann plot showing the ratio of DS DNA yield estimates by drop spectrophotometry (NanoDrop, red) or fluorometry (Qubit, blue), relative to qPCR.** Dots are separated along the horizontal axis on the basis of absolute DNA yield as estimated by qPCR. Dense dots represent samples with high (>1.6) $A_{260}/A_{280}$ ratios; pale dots represent low $A_{260}/A_{280}$ ratios.

Table 7. Correlation coefficients for DNA measurements. Values<0.3 reflect little to no linear relationship between the two methods. Values>0.8 are marked with bold type and indicate high covariance among obtained quantifications. Qubit: Fluorometry for DS DNA; Nano SS: drop spectrophotometry with settings for SS DNA; Nano DS, drop spectrophotometry with settings for DS DNA, gel: agarose gel electrophoresis stained with SYBR green. Orange fill indicates that the measurements are considered appropriate for the material (i.e., measuring DS DNA concentration of where DS DNA is expected or SS DNA concentration where SS DNA is expected).

| | All samples | Ammonia | | | Qiagen | | |
|---|---|---|---|---|---|---|---|
| | | All | Intact | Homogenised | All | QBT | QMK |
| qPCR vs Qubit | 0.65 | **0.87** | **0.78** | **0.92** | 0.49 | 0.46 | 0.64 |
| qPCR vs nanoSS | 0.11 | 0.03 | 0.13 | -0.003 | 0.24 | 0.05 | 0.43 |
| qPCR vs nanoDS | 0.46 | 0.52 | 0.79 | 0.33 | 0.20 | 0.15 | 0.26 |
| qPCR vs gel | 0.62 | 0.70 | **0.88** | 0.62 | 0.41 | 0.34 | **0.83** |
| Qubit vs nanoSS | -0.05 | 0.05 | 0.08 | 0.05 | 0.27 | 0.13 | 0.44 |
| Qubit vs nanoDS | 0.28 | 0.60 | **0.92** | 0.41 | 0.33 | 0.39 | 0.26 |
| Qubit vs gel | 0.52 | **0.84** | **0.94** | **0.82** | 0.20 | -0.02 | 0.78 |
| nanoSS vs nanoDS | 0.20 | 0.06 | 0.01 | 0.03 | 0.20 | 0.18 | 0.22 |
| nanoSS vs gel | 0.08 | -0.06 | -0.11 | 0.06 | 0.07 | -0.09 | 0.55 |
| nanoDS vs gel | 0.54 | 0.66 | **0.97** | 0.21 | 0.17 | 0.08 | 0.48 |

Table 8. Positive samples for *Neoehrlichia mikurensis* (NM) and *Borrelia burgdorferi* s.l. (Bb) by method, detected by qPCR.

| Method | Analysed | NM | Bb |
|---|---|---|---|
| AC | 50 | 4 (2M, 2N) | 10 (1F, 1M, 8N) |
| ANC | 50 | 2 (2F) | 5 (2F, 2M, 1N) |
| QBT | 50 | 2 (1M, 1N) | 6 (1F, 5N) |
| QMK | 50 | 2 (1M, 1N) | 5 (2F, 1M, 2N) |

Table 9. Positive samples for *Neoehrlichia mikurensis* (NM) and *Borrelia burgdorferi* s.l. (Bb) by locality, detected by qPCR.

| Locality | Analysed | NM | Bb | CI |
|---|---|---|---|---|
| Jomfruland | 161 (1M, 160N) | 3 (3N) | 16 (1M, 15N) | 3 (3N) |
| Langøya | 19 (10F, 5M, 4N) | 3 (2F, 1M) | 8 (5F, 3M) | 1 (1F) |
| Asdal | 20 (2F, 6M, 12N) | 4 (3M, 1N) | 2 (1F, 1N) | 0 |
| Total | 200 (12F, 11M, 177N) | 10 (2F, 4M, 4N) | 26 (6F, 4M, 16N) | 4 (1F, 3N) |

F: females; M: males; N: nymphs; CI= Coinfected.

### *Neoehrlichia mikurensis* and *Borrelia burgdorferi* s.l. detection

*Neoehrlichia mikurensis* and *B. burgdorferi* s.l. were detected in 10 (5%) and 26 (13%) ticks, respectively. All four extraction methods yielded positive samples (Table 8). The prevalence of *B. burgdorferi* s.l. was 42% on Langøya and 10% at Jomfruland and Asdal. The prevalence of *N. mikurensis* was 20% in Asdal, 16% on Langøya and 2% on Jomfruland. Four samples were coinfected (Table 9).

## Discussion

In this study we compare the yield and quality of DNA obtained with different methods from sub-optimally stored ticks. Samples were originally stored immersed in absolute ethanol, but after 11 years of storage at 4°C, it had evaporated in almost all samples. Comparisons were made by using two commercially available silica membrane-based methods and two variants of the in-house ammonium hydroxide extraction method using either intact or homogenized ticks. Yields,

based on qPCR, were similar for all four methods, but very variable, which probably reflects the poor state of preservation of the ticks.

One reason for exploring the ammonium hydroxide methods was the possibility of minimising DNA loss, since it was expected that, due to the storage conditions, much of the DNA would be in the form of small fragments that might not be well retained by the silica membranes used in the Qiagen kits. Although gel electrophoresis does suggest that more low-molecular weight DNA is recovered by the ammonium hydroxide methods, this is not reflected in significantly higher DNA yields as judged by qPCR.

All methods yielded amplifiable DNA and *B. burgdorferi* and/or *N. mikurensis* was detected in samples extracted by all four methods. A slight-to-moderate degree of PCR inhibition was observed in a minority (18%) of ammonium-hydroxide-extracted homogenized ticks. All other samples were non-inhibitory. Judging by $A_{260}/A_{280}$ ratios, none of the methods yielded consistently pure DNA, which may account for difficulties in obtaining consistent yield estimates, as described below. The Qiagen Blood and Tissue kit yielded the purest DNA, but the mean $A_{260}/A_{280}$ ratio of 1.6 was still well below the target value of 1.8–2.0. The difference relative to the Qiagen Mini Kit is unexpected as the reagents in the two kits are identical (Danuta Kaczmarzyk, Qiagen technical support, pers comm.) and the only difference between the proto-cols was the temperature for a ten-minute post-lysis incubation; 56°C for the Blood and Tissue Kit and 70°C for the Mini Kit. The ammonium hydroxide extraction methods involve no cleanup at all and are not expected to yield pure DNA.

Ammonium hydroxide extracts and Qiagen extracts showed different electrophoresis patterns. Qiagen extracts showed a tight peak of high-molecular-weight (>15 Kb) DNA with a tail of lower molecular material, a pattern typical of genomic DS DNA extracts. The high molecular weight peak may be explained by the sheering effects of pipetting and homogenization by bead-beating and/or selective loss of low-molecular-weight fragments during the extraction procedure, together with the low resolution of 4% agarose gels at high molecular weight. Twelve Qiagen extracts showed a flat distribution which may indicate a limited degree of degradation. The ammonia extracts showed a flat distribution, beginning at a higher apparent molecular weight and extending to ca. 1 Kb. The presence of low molecular weight material is not unexpected; the ammonia-based extraction procedure offers no opportunity for size fractionation and the DNA is denatured, which will result in the fragmentation of nicked DNA. The high molecular weight material might be attributed to lower degrees of sample handling resulting in less shearing of DNA in ammonia extracts, but in that case, less high molecular weight material would be expected in homogenised ticks, which we did not observe. We attribute this rather to the formation of complexes of incorrectly renatured SS DNA fragments, as the method, which involves heating to 99°C under denaturing conditions followed by rapid cooling to 4°C, is conducive to incorrect renaturation [23].

Although some studies report poor amplification with ammonia extracts [24], others, in common with this study, report satisfactory results [25]. The difference might be explained by our observation, using pH indicator dyes, that the pH of samples only returns to neutral after several days' storage. Use of samples immediately after extraction would therefore be likely to give unsatisfactory results, as the pH might exceed the optimum for Taq polymerase [26]. Also, homogeniza-tion of the samples by bead-beating resulted in some inhibitory samples, which has been previously observed for the Qia-gen Blood and Tissue kit [25]. Thus, this extra step is counterproductive. Bisection of ticks instead of bead-beading might be a good alternative since it will open the exoskeleton without dispersing inhibitory substances [25]. However, it may not be easily applicable when a large number of samples is required as in epidemiological surveys.

Although we used multiple DNA concentration measurements in the expectation of achieving more robust results, our results instead served to highlight the difficulty of obtaining reliable estimates of DNA concentration from samples of this kind. Correlation between the measures was generally poor; correlation coefficients varied between -0.11 and 0.97 without any noticeable pattern or logic; in particular, there was no evidence that correlation coefficients were higher when the mea-surement method used was appropriate for the DS or SS state of the DNA, in fact, rather the contrary. Combined with the low purity indicated by $A_{260}/A_{280}$ measurements, this suggests that much of the signal in our DNA measurements is due to impurities in the sample rather than DNA itself. Thus, we chose to use qPCR as our gold-standard method, since, although

it is theoretically the least precise of the methods used, it is immune to spurious positive results caused by non-DNA contaminants. Relative to qPCR, drop spectrophotometry overestimated DNA concentration, while fluorometry and gel electrophoresis underestimated it across all methods, with the exception of the Qiagen Blood and Tissue kit extracts, where, in general, underestimation by fluorometry was not observed. For the ammonium hydroxide extracts, this is expected, as they were denatured and therefore contained predominantly SS DNA. SS DNA has a higher absorbance at 260nm, while it does not fluoresce with the dye used in the Qubit kit. However, the overestimate persisted when spectrophotometric settings for SS DNA were used. Furthermore, drop spectrophotometry also overestimated DNA yields for the DS Qiagen extracts and fluorometry underestimated values for Qiagen Mini kit extracts. Earlier studies report a tendency for drop spectrophotometry to overestimate low DNA concentrations [5,27] and for Qubit fluorometry to underestimate them [6].

Our results present a dilemma for the researcher who wishes to get the best out of a set of suboptimally stored tick samples. As indicated by the wide variation of yield seen in this study, many of the samples will have lost 99% or more of their DNA. This will compromise pathogen detection and lead to underestimation of pathogen prevalence. In such a situation, accurate DNA concentration measurements are essential in order to appropriately screen out degraded samples and/or adjust sample volumes in PCR tests. However, as our results show, neither easily implemented, low cost spectrophotometry [28] nor more sophisticated fluorometric assays are adequate for this purpose.

One solution to this problem would be to apply more elaborate purification procedures in order to obtain samples pure enough to allow an accurate measurement of DNA concentration, or at least, a trustworthy null value. Such an approach will inevitably entail increased labor and material costs. The alternative would be to use qPCR for quantification. Although it has been suggested that qPCR may not be cost-effective [29,30], our results suggest that the extra expense can be counterbalanced by applying ammonia extraction of intact ticks, which is essentially cost-free and can replace more expensive kit-based methods without compromising yield or amplifiability. Relative to the cost of kit-based extraction (approximately $5 per sample), qPCR is cheap (approximately $1 per sample), the more so if it were multiplexed with the pathogen detection test.

Our results therefore suggest that the way to get the best out of a bad sample – at least where the sample is a suboptimally stored questing tick and the analysis method is PCR – is to extract it intact with ammonia, check the DNA yield by qPCR and exclude samples or adjust input volumes accordingly. It should be noted, however, that this approach is unlikely to be suitable for engorged ticks, as the haem component of blood, which is a known PCR inhibitor [31] will not be removed by ammonia extraction, although the small amounts of the blood meal reported to be retained through ecdysis [32] do not appear to present a problem in this study. Neither would we expect this method to be suitable for demanding downstream analyses, such as high-throughput sequencing. Finally, since the DNA is denatured, it would be inapplicable to any method requiring DS DNA.

*Neoehrlichia mikurensis* and *B. burgdorferi* s.l. were targeted as a way of validating the possibility of detecting bacterial DNA by the four extraction methods. *N. mikurensis* is an emerging pathogen and *B. burgdorferi* is the most common tick-borne bacteria in the northern hemisphere [33]. Our results add one new location, Asdal, to the sites in southeastern Norway where *B. burgdorferi* s.l. (10%) and *N. mikurensis* (20%) are known to occur. For Langøya, the prevalence of *B. burgdorferi* s.l. (42%), and *N. mikurensis* (15%), are higher than previously reported [34]; for Jomfruland, the prevalence of *B. burgdorferi* s.l. (10%) is close to that previously reported in questing ticks [35], while the prevalence of *N. mikurensis* (2%) is lower than the 5% previously reported [22]. As the prevalences reported here are affected by small sample size and/or poor sample quality, we consider them imprecise and potentially underestimated.

## Conclusion

This study shows that ammonium hydroxide lysis and silica-membrane based kits perform equally well for DNA extraction from sub-optimally stored *I. ricinus* ticks. Homogenisation of the ticks using steel beads prior to ammonia extraction caused inhibition and did not improve yield. Therefore, despite the notorious toughness of the tick exoskeleton, it is apparently permeable to DNA after ammonium hydroxide treatment. Yields varied by more than a thousandfold, presumably

reflecting the impact of suboptimal storage on DNA preservation and underlining the need for accurate DNA quantification to prevent inaccurate prevalence estimates. However, due to the impurity of the samples and low DNA concentrations, reliable quantification could not be achieved by convenient low-cost methods such as drop spectrophotometry or fluorometry, leaving qPCR as the only viable alternative.

We conclude that ammonium hydroxide extraction of intact ticks, which involves a minimal sample manipulation and has a negligible cost, combined with DNA quantification by qPCR is the most labour- and cost-effective method of processing poorly preserved ticks.

The highly variable DNA yields in this study led to noisy data, and it would be worth repeating it with optimally stored ticks in order to better compare the performance of the methods.

## Supporting information

**S1 Fig. Agarose gel electrophoresis 1.** From well 1–10: AC-N, AC-N, AC-N, AC-N, AC-N, QMK-N, QMK-N, QMK-N, QMK-N, QBT-N.
(JPG)

**S2 Fig. Agarose gel electrophoresis 2.** From well 1–10: QMK-N, QMK-N, QBT-N, QMK-N, QBT-N, QBT-N, QBT-N, QBT-N, QBT-N, ANC-N.
(JPG)

**S3 Fig. Agarose gel electrophoresis 3.** From well 1–10: QMK-N, ANC-N, QMK-N, QBT-N, QBT-N, ANC-N, AC-F, ANC-N, ANC-N, QMK-N.
(JPG)

**S4 Fig. Agarose gel electrophoresis 4.** From well 1–10: ANC-N, QBT-N, ANC-N, ANC-N, ANC-N, QMK-N, QMK-N, ANC-N, ANC-M, QMK-N.
(JPG)

**S5 Fig. Agarose gel electrophoresis 5.** From well 1–10: AC-M, QBT-N, QBT-N, ANC-N, AC-N, ANC-N, QBT-N, QBT-N, ANC-N, QBT-N.
(JPG)

**S6 Fig. Agarose gel electrophoresis 6.** From well 1–10: ANC-F, QBT-N, ANC-N, QBT-F, QMK-N, ANC-N, AC-N, ANC-N, ANC-N, QMK-N.
(JPG)

**S7 Fig. Agarose gel electrophoresis 7.** From well 1–10: AC-F, AC-F, ANC-F, ANC-F, QBT-N, QMK-N, QBT-N, QBT-N, QBT-N, QBT-N.
(JPG)

**S8 Fig. Agarose gel electrophoresis 8.** From well 1–10: ANC-N, QBT-N, ANC-N, QMK-N, ANC-N, ANC-N, AC-N, QBT-N, QBT-F, AC-N.
(JPG)

**S9 Fig. Agarose gel electrophoresis 9.** From well 1–10: ANC-N, AC-N, QMK-N, ANC-N, ANC-N, QBT-N, ANC-N, ANC-N, ANC-N, AC-N.
(JPG)

**S10 Fig. Agarose gel electrophoresis 10.** From well 1–10: QBT-N, ANC-N, AC-N, QMK-N, QMK-N, QBT-N, AC-N, QBT-N, QBT-N, AC-N.
(JPG)

**S11 Fig. Agarose gel electrophoresis 11.** From well 1–10: ANC-N, QMK-N, AC-N, ANC-N, QMK-N, AC-N, QMK-N, AC-N, ANC-N, QMK-N.
(JPG)

**S12 Fig. Agarose gel electrophoresis 12.** From well 1–10: AC-N, AC-N, QMK-N, QBT-N, ANC-N, AC-N, QBT-N, QMK-N, AC-N, QBT-N.
(JPG)

**S13 Fig. Agarose gel electrophoresis 13.** From well 1–10: QMK-N, AC-N, ANC-M, ANC-N, ANC-N, AC-N, QBT-N, QBT-N, ANC-N, QMK-F.
(JPG)

**S14 Fig. Agarose gel electrophoresis 14.** From well 1–10: QBT-N, QMK-N, QMK-F, AC-N, ANC-N, QMK-N, QMK-N, QBT-F, AC-N, QBT-N.
(JPG)

**S15 Fig. Agarose gel electrophoresis 15.** From well 1–10: AC-N, QBT-N, AC-N, AC-N, ANC-N, QBT-N, QMK-N, QBT-N, QMK-N, QBT-N.
(JPG)

**S16 Fig. Agarose gel electrophoresis 16.** From well 1–10: ANC-N, AC-N, AC-N, QMK-N, QBT-N, ANC-N, QMK-N, AC-N, QMK-N, AC-N.
(JPG)

**S17 Fig. Agarose gel electrophoresis 17.** From well 1–10: AC-N, AC-N, QMK-F, AC-N, AC-M, AC-N, QMK-N, AC-N, AC-N, QMK-N.
(JPG)

**S18 Fig. Agarose gel electrophoresis 18.** From well 1–10: AC-N, ANC-N, ANC-N, ANC-N, ANC-M, AC-N, QMK-N, QMK-N, AC-N, ANC-N.
(JPG)

**S19 Fig. Agarose gel electrophoresis 19.** From well 1–10: QBT-N, QMK-N, QBT-N, QMK-N, QMK-N, QMK-N, AC-N, AC-N, QMK-M, QMK-N.
(JPG)

**S20 Fig. Agarose gel electrophoresis 20.** From well 1–10: QBT-M, AC-M, ANC-N, AC-N, QBT-N, QBT-N, QBT-N, QMK-M, QMK-M, QBT-M.
(JPG)

**S21 Fig. Comparison between the four extraction methods based on the A260 nm absorbance values by NanoDrop.** Results from the Kruskal-Wallis test are shown and significant ($p < 0.05$) differences in pair-wise comparison by Wilcoxon-Mann-Whitney with Bonferroni adjustment are shown at the top.
(PDF)

**S22 Fig. Comparison between the four extraction methods based on the amplification of the *Ixodes ricinus 5.8S* Ct values.** Results from the Kruskal-Wallis test are shown and significant ($p < 0.05$) differences in pair-wise comparison by Wilcoxon-Mann-Whitney with Bonferroni adjustment are shown at the top.
(PDF)

**S1 Table. Overall results from the four extraction methods.**
(DOCX)

**S2 Table. Raw data.** Qubit, Nano_DS, Nano_SS, IxodesQ_ng and Gel_ng show the estimated DNA quantity in ng obtained by each measurement method. SpikeQ and Neo_spikeCt show the genome copies number and Ct values, respectively, obtained from the inhibition test. Qubit_PCR_Ratio and Nano_PCR_Ratio show the ratio obtained from Qubit and qPCR estimations and NanoDrop and qPCR estimations, respectively.
(CSV)

## Acknowledgments

The authors thank Dag Hvidsten for providing the samples used for this study, Rose Vikse for her help with the manuscript, Trond Leirstang of the USN print shop for converting the gel images, and the reviewers for their valuable contribution to the quality of this document.

## Author contributions

**Conceptualization:** Andrea P. Cotes-Perdomo, Kristian Alfsnes, Åshild K. Andreassen, Andrew Jenkins.

**Data curation:** Andrea P. Cotes-Perdomo, Kenedith Méndez-Gutierrez, Andrew Jenkins.

**Formal analysis:** Andrea P. Cotes-Perdomo, Kenedith Méndez-Gutierrez, Andrew Jenkins.

**Funding acquisition:** Kristian Alfsnes, Åshild K. Andreassen.

**Methodology:** Andrea P. Cotes-Perdomo, Kenedith Méndez-Gutierrez, Andrew Jenkins.

**Software:** Kenedith Méndez-Gutierrez.

**Supervision:** Andrew Jenkins.

**Visualization:** Andrea P. Cotes-Perdomo, Kenedith Méndez-Gutierrez.

**Writing – original draft:** Andrea P. Cotes-Perdomo, Andrew Jenkins.

**Writing – review & editing:** Andrea P. Cotes-Perdomo, Kristian Alfsnes, Åshild K. Andreassen, Andrew Jenkins.

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
