## [Decision Letter · Decision Letter 0]

6 Jan 2025

PONE-D-24-53767Making the best of a bad sample: Comparison of DNA extraction and quantification methods using sub-optimally stored Ixodes ricinus ticksPLOS ONE

Dear Dr. Cotes,

Thank you for submitting your manuscript to PLOS ONE. After careful consideration, we feel that it has merit but does not fully meet PLOS ONE’s publication criteria as it currently stands. Therefore, we invite you to submit a revised version of the manuscript that addresses all of the points raised during the review process, including requested experimental details, etc.

We look forward to receiving your revised manuscript.

Kind regards,

Brian Stevenson, Ph.D.

Academic Editor

PLOS ONE

“This work was in part financed by The Norwegian Lyme Borrelioses society (NLBF).”

Reviewers' comments:

Reviewer's Responses to Questions

**Comments to the Author**

1. Is the manuscript technically sound, and do the data support the conclusions?

Reviewer #1: Yes

Reviewer #2: No

2. Has the statistical analysis been performed appropriately and rigorously? 

Reviewer #1: Yes

Reviewer #2: No

3. Have the authors made all data underlying the findings in their manuscript fully available?

Reviewer #1: Yes

Reviewer #2: No

4. Is the manuscript presented in an intelligible fashion and written in standard English?

Reviewer #1: Yes

Reviewer #2: Yes

5. Review Comments to the Author

Reviewer #1: After review of this manuscript, I recommend that it be accepted for publication with minor revision.

The purpose of the manuscript is to detail a study comparing DNA extraction methods of several poorly stored tick DNA samples. The methods of extraction used included ammonia hydrolysis of both homogenized and intact ticks, and two commercial silica membrane-based kits. The authors keep a tight focus on this objective throughout the paper, which makes it an efficient and informational read.

The abstract is well written and to the point. It sufficiently details the purpose of the study, brief methodology and results, and the primary conclusion; I appreciate input of final sentence of the abstract: “Our findings indicate that for the purposes of qPCR analysis, ammonia hydrolysis of intact ticks, a very cheap and simple method is as good as any of the other methods tested.”. It is refreshing to see an abstract written in the intended format, instead of a short rambling of the entire paper in summation.

Minor notes that I feel would strengthen the paper are listed below:

- Introduction, line 45-46: What about these conditions (Storage in ethanol) increase DNA quality? Is the quality comparable to some of the storage solutions available on the market?

- Introduction, line 60: Remove the “.” After the citation.

- Methods, lines 121-131: Is the method of extraction using ammonia the standard?

- Methods, line 133 and 134-136: Why were these exceptions made?

- Methods, line 175: Missing “.” at the end of the sentence.

- Methods, line 186: De-underline the “.”.

- Methods, line 190: Abbreviate Neoehrlichia following the first mention in the paper.

- Results, line 196: Spell out the scientific name if it is the first word of the sentence.

- Discussion, line 320: Abbreviate Borrelia and Neoehrlichia following the first mention in the paper, as long as it is not the first word of the sentence.

- Discussion, line 325: Please describe more in detail, if known, about why the Qiagen Blood and Tissue kit yielded the purest DNA.

- Discussion: line 349-350: Why is this data not included in the study? Is it included in a separate manuscript? If so, please cite it. If not, is it possible to include it in this study?

- Discussion, line 362: qPCR not QPCR.

- Discussion, line 370: change overestimate to overestimated.

- Discussion, line 401-402: Abbreviate Borrelia and Neoehrlichia following the first mention in the paper, as long as it is not the first word of the sentence.

Reviewer #2: Cotes-Perdomo et al evaluate different methods of DNA extraction and quantification of Ixodes ricinus ticks that were stored for over 10 years at room temperature in 100% ethanol, which had evaporated from several samples prior to extraction. The authors tested ammonia hydrolysis methods (with homogenized ticks and non-homogenized ticks) and two industry standard kits from Qiagen (with homogenized ticks) to extract DNA. They also tested different DNA quantification methods (qPCR, fluorometry, drop spectrophotometry, gel electrophoresis) after DNA extraction to determine the best methods to measure quality and quantity. The authors found that extracting DNA from whole ticks using the ammonia hydrolysis method was not only better at extracting more DNA, but was also more cost-effective and less labor-intensive than the other methods. Further, qPCR was the best method to quantify DNA.

Overall, the authors provide good information on a question that likely many authors in the tick world have had about the viability of samples, especially if resources for DNA extraction and/or testing are not readily available. To improve the manuscript, there are several major changes and clarifications that should take place before the paper is ready for publication.

The major comments and line-by-line comments are provided in the attached document.

6. PLOS authors have the option to publish the peer review history of their article (what does this mean? ). If published, this will include your full peer review and any attached files.

**Do you want your identity to be public for this peer review?** For information about this choice, including consent withdrawal, please see our Privacy Policy .

Reviewer #1: No

Reviewer #2: No

---

## [Author Response · Author response to Decision Letter 1]

21 Mar 2025

Answer to reviewers

The authors are very grateful to both reviewers for their comments and suggestions, their work has proven extremely valuable for the quality of the manuscript.

Journal requirements of style and file formatting has been applied to figures and the manuscript. Ethical statement has been included in the methods section and the role of the founders was clarified in the cover letter. The uncropped pictures of electrophoresis agarose gels are being supplied as supplementary material as well as the data set use for the analysis.

Answers to both reviewers’ comments are provided below immediately after each comment in italics.

Reviewer 1

- Introduction, line 45-46: What about these conditions (Storage in ethanol) increase DNA quality? Is the quality comparable to some of the storage solutions available on the market? – We are not aware of any use of other preservatives for ticks.

- Introduction, line 60: Remove the “.” After the citation. – “.” Was deleted.

- Methods, lines 121-131: Is the method of extraction using ammonia the standard? – There is to the best of our knowledge no standard method for extracting DNA from ticks. We have place the QIAGEN methods first, in response another request, indicating that they are the more frequently used methods.

- Methods, line 133 and 134-136: Why were these exceptions made? – First exception, in the previous paragraph, the fluid was ammonia; second exception, given the nature of the hard tick’s cuticle, the lysis is normally extended to allow the release of nucleic acids.

- Methods, line 175: Missing “.” at the end of the sentence. - “.” Was included.

- Methods, line 186: De-underline the “.”. - “.” De-underlined.

- Methods, line 190: Abbreviate Neoehrlichia following the first mention in the paper. – Checked and modify along the document.

- Results, line 296: Spell out the scientific name if it is the first word of the sentence. - Checked and corrected.

- Discussion, line 320: Abbreviate Borrelia and Neoehrlichia following the first mention in the paper, as long as it is not the first word of the sentence. – Checked and modify along the document.

- Discussion, line 325: Please describe more in detail, if known, about why the Qiagen Blood and Tissue kit yielded the purest DNA. After consulting with QIAGEN technical support, it transpires that there is no difference between the two QIAGEN kits used except for the temperature of a 10 minutes post-lysis incubation in ‘Buffer AL’ (56°C vs 70°C). This makes the difference in purity difficult to account for, and we have modified the text accordingly.

- Discussion: line 349-350: Why is this data not included in the study? Is it included in a separate manuscript? If so, please cite it. If not, is it possible to include it in this study? We have added a description of this experiment. See section 2.2 in the methods and section 3.1 in the results.

- Discussion, line 362: qPCR not QPCR. - Corrected

- Discussion, line 370: change overestimate to overestimated. – Changed.

- Discussion, line 401-402: Abbreviate Borrelia and Neoehrlichia following the first mention in the paper, as long as it is not the first word of the sentence. – Done.

Reviewer 2

Major comments

One major concern with the paper is that the results are not separated by tick life stage. The authors tested nymphal, female, and male ticks, all of which may have different amounts of DNA present. This could potentially affect the statistics of the study and thusly, interpretation of the results. If the authors chose to combine the life stages during the data analysis phase due to no significant difference in the amount of DNA present across the different life stages and the data were not shown, then they should mention this circumstance and provide the data either in the manuscript or as supplementary materials. Separating the results based on life stage would be more impactful as many researchers often selectively test different life stages due to resource availability, pathogen presence in certain life stages, etc.

- No statistical differences were observed between methods when samples were separated by life stage and sex, supplementary table 1 contains an overview or the results separated by life stage and sex by method. Data will be submitted as supplementary material separated by life stage. Additionally, the whole data set has been included in the supplementary material.

Furthermore, the methods need additional clarifying details, especially the statistical analyses. The authors reported results for which no methods were mentioned, or the methods did not provide enough detail to clearly interpret the results. For example, the authors did not add information on the ratios they calculated to compare the DNA extraction and measurement methods; the results were explained in lines 266-274 and table 5. More specifically, what values were used to calculate the ratios? In addition, their GLM also needs more context, including information on the covariates that went into the model and the outcome variable as well as the link function and distributions used. It is difficult to interpret the results mentioned in lines 194-196 and 291-294 without the context of what went into the model. The authors also conducted a correlation analysis and produced correlation coefficients to compare the DNA measurement across the DNA extraction methods (see lines 279-290, table 6), but this was not mentioned anywhere in the methods. Information on how the coefficients were calculated (what values were used?) and how the authors interpreted the results would be helpful. While some of the methods were provided in the figure caption, the information was not in the actual main manuscript text. With a lot of missing information from the methods, the results are difficult to interpret.

- The settings and interpretation for the GLM are now clarified in the methods and results sections. Originally only one model was used, we have now done a model selection to find the more parsimonious one based on AIC, it does not change the results.

- Correlation coefficients, and the calculation of ratios for the Bland-Altmann plot and table 5 has been clarified.

Furthermore, there was no information on the role of testing for N. mikurensis and B. burgdorferi in the study. While N. mikurensis was used to detect PCR inhibition, this was not the case for B. burgdorferi. It’s reasonable for the authors to include those results in the manuscript, but that should be clearly stated as a new objective in the study because testing for those two pathogens seemed out of place when that was not the main purpose of this study. If the authors choose to keep this information in the manuscript, they need to include the significance of these pathogens in the introduction and/or discussion to justify reporting the results in the manuscript.

- The inhibition test has been left alone as one section, followed by a section specific for pathogen detection. This has been also included in the introduction as one of the aims of the study (being able to detect naturally acquired bacteria from this material).

Finally, the figures and tables could be improved by providing more context in the captions and/or the manuscript text. Similar to what was said previously about results reporting, it would help readers if the authors included how the data in the figures/tables were calculated, where necessary. Figure 2 doesn’t really add much to the manuscript. Instead, I suggest having a figure show the different shapes described in lines 239-252. Figure 3 could be improved if the authors included all of the DNA quantification methods in one figure to compare DNA yields per DNA extraction and quantification method. That is, take supplementary figures 21-23 and make a multi-panel figure so that readers can compare the quantities directly rather than having to pull up the supplementary figures.

- A multi-panel figure has been included as suggested. Clarifications has been made about how the data was calculated for figures, tables and statistical analysis. Figure 2 has been replaced with a figure summary of the shapes described in the text.

Line-by-Line Comments

Abstract: I would check the submission instructions for the journal, but the abstract should probably be one large paragraph rather than separate paragraphs. – non specified in the instruction from the journal, nevertheless, the paragraphs were unified.

Lines 15-19: It would be helpful to briefly include how performance was evaluated for each DNA extraction method and how yield and purity were evaluated. This would provide more context overall to the rest of the abstract. – Information was included in the first paragraph of the abstract.

Line 16: Which life stages of ticks were tested? - Information was included in the first paragraph of the abstract.

Line 20 (and elsewhere): There are several instances where the authors use adverbs that may not necessarily be useful in the sentence’s context and/or cannot be quantified. In this case, “very” does not need to be included. The authors should go through and remove these adverbs from the manuscript accordingly. – noted and corrected.

Line 25 (and elsewhere): “… drop spectrophotometry…” – Corrected.

Line 26: What do you mean by “correlated poorly?” What values were compared for the correlation and how were the correlations calculated? – Clarification was added

Lines 28-29: Which methods were used to detect N. mikurensis and B. burgdorferi? – Information included.

Introduction

Line 60: “… hydrolysis method (11-13) has the advantage…” – done.

Line 60 (and elsewhere): Is it “ammonia hydroxide hydrolysis” or “ammonia hydrolysis?” There are several instances where either of the terms are used, but it would be better to stick with one term and to use it consistently throughout the manuscript. – Changed.

Line 71: “… for pure double-stranded (DS) or single-stranded (SS) DNA, and corrections…” - Change has been done.

Lines 70-78: The sentences need to be restructured as they are quite long-winded and confusing to read. I suggest something like this – “While this method is quite accurate for pure double-stranded (DS) or single-stranded (SS) DNA and corrections for the presence of moderate amounts of protein are available, A260 will not be an accurate measure of DNA concentration if the sample contains a mixture of DS and SS DNA (whose A260s differ) or if other substances containing aromatic rings, such as ATP, RNA, and phenol are present. Further, the A260:280 ratio is not a foolproof measure of molecular purity as many impurities have no absorbance in this region. It would be possible to construct a mixture of aromatic compounds having an A260:280 ratio of 1.8 while containing no DNA at all, for example.” This is also where you have the first mention of double-stranded and single-stranded DNA, so only the abbreviations can now be used throughout the rest of the manuscript. – Suggestions applied.

Lines 101-102: “… two related commercial DNA extraction kits that use silica membranes…” – Suggestion applied.

Line 104 (and 392): It would be helpful to explicitly define “sub-optimal” conditions either here in this line, methods, or both. This was alluded to earlier in the introduction (lines 44-47), but it should be more clearly defined. – Sub-optimal has been clarified in the introduction and discussion.

Methods

Line 113: Could ticks still submerged in ethanol have more DNA preserved? Did any samples have ticks still submerged in ethanol when it was time to conduct DNA extractions or were all considered dry? If submerged ticks were used, this could have affected the results of the DNA extraction and quantities. Please provide more information, maybe an approximate percentage of ticks that were still in ethanol prior to the extractions- ~90% were dry, the still submerged or partially submerged ticks were randomly extracted with the four methods. Information has been included in the manuscript.

Lines 121-131: Please provide citations for the extraction methods using ammonia. – Citation provided.

Line 139 (and elsewhere): Should “concentration” be replaced with “quantity?” In most cases, concentrations are not the same as quantities. - We have reviewed the manuscript and replaced concentration with amount or yield where appropriate. However, instrument measurements reported concentration in ng/µl and in these cases, we do not feel it would be appropriate to substitute 'yield'.

Lines 139-177: The order of the methods to quantify and evaluate DNA should be the same order in which they are introduced in the introduction. Similarly, the results should follow a consistent order as well, where appropriate. In addition, please provide the amounts of extracted DNA used for each DNA quantification method. – The order of the methods used to quantify and evaluate DNA has been modified as suggested. The amount of DNA employed in each test has also been included in the text.

Lines 140-142: Please add more information on how the gel electrophoresis was run. Include information such as the voltages used, amount of sample used per well (see previous comment) and time used to run the gels. – Information included.

Lines 163-165 and Tables 2 and 3: Tables should be switched, where the table of primers is listed first and then the PCR settings. In addition, a brief description about 5.8s rRNA should be provided (what is it? What does it control?). A reference from the literature describing 5.8S rRNA, its primers, and protocol should also be provided in the main text and not just in the table. – References have been included in the main text and a brief description of 5.8S has been provided.

Lines 172-175: “… as an anomalous amplification curve.” Also, please provide a reference for this sentence. – Sentence has been re-written, and a reference provided.

Lines 178-186: This section seems very out of place compared to the rest of the manuscript. I understand the purpose of including N. mikurensis as a way to test for PCR inhibition, but then why include B. burgdorferi if that was not going to be used in the same way as N. mikurensis? – Inhibition test and pathogen detection have been divided into two different sections.

Line 183: Were all samples spiked with pNeo? Or just some? If only some were spiked, how were they selected? – All samples were spiked, clarification included in text.

Lines 183-186: Can you provide more information on PCR inhibition and how it relates to sample spiking in the introduction and/or briefly in the methods? - We have added a description of the spiking method and provided references in the introduction.

Line 186: Remove the underline from the period at the end of the sentence. – Done.

Lines 188-193: The way the sentence is currently written, it sounds like you are comparing the listed measurements to each other rather than comparing the measurements of the different extraction methods. If the latter is the case, please rephrase the sentence for clarity. In addition, “results are presented in Supplementary Table 1 and corresponding figures” can be deleted. This can be mentioned in the actual results section. Furthermore, supplementary table 1 can probably be moved to the main text as this is interesting information. If it’s moved to the main text, you should bold or add notations that mark significant differences. – The sentence has been re written. Supplementary table 1 was moved to the main manuscript and it is now Table 5. It is referred in the text together with Figure 2 (multi-panel now) to show the statistical support estimated by Kruskal-Wallis and pairwise comparisons by Mann-Whitney-Wilcoxon tests for a better readability.

Line 194: “A generalized linear model…” – Done.

Lines 194-196: This sentence does not make sense. More information on the GLM needs to be provided in general, see major comments. – Information about the GLM has been provided.

Line 198: R was used to do the statistics, but please also provide R packages that were used to calculate any measurements or produce data. In addition, information on Excel was not provided here even though a function used in Excel was mentioned in a table to calculate correlation coefficients (lines 288-289). Information on how the correlation coefficients were calculated were also not pro

---

## [Editor Report · Decision Letter 1]

6 Apr 2025

Making the best of a bad sample: Comparison of DNA extraction and quantification methods using sub-optimally stored Ixodes ricinus ticks

PONE-D-24-53767R1

Dear Dr. Cotes,

We’re pleased to inform you that your manuscript has been judged scientifically suitable for publication and will be formally accepted for publication once it meets all outstanding technical requirements.

Kind regards,

Brian Stevenson, Ph.D.

Academic Editor

PLOS ONE
---

## [Editor Report · Acceptance letter]

PONE-D-24-53767R1

PLOS ONE

Dear Dr. Cotes-Perdomo,

I'm pleased to inform you that your manuscript has been deemed suitable for publication in PLOS ONE. Congratulations! Your manuscript is now being handed over to our production team.

Kind regards,

on behalf of

Prof. Brian Stevenson

Academic Editor

PLOS ONE